# Efficient Isolation of Circulating Tumor Cells Using Ultra-Convenient Substrates Based on Self-Assembled Hollow MnO$_2$ Nanoparticles

**Rui Li** [1,2,3,*], **Yuankun Wang** [1] **and Tengfei Long** [1]

1 Xinjiang Key Laboratory of Solid State Physics and Devices, Xinjiang University, Urumqi 830046, China
2 School of Physics Science and Technology, Xinjiang University, Urumqi 830046, China
3 Key Laboratory of Artificial Micro- and Nano-Structures of Ministry of Education, School of Physics and Technology, Wuhan University, Wuhan 430072, China
* Correspondence: lirui@xju.edu.cn

**Abstract:** An efficient and active sorting platform of circulating tumor cells (CTCs) is still a challenge in clinical research. In this paper, we design a novel system based on hollow MnO$_2$ nanoparticles for the capture and release of CTCs. Using the self-assembly method, we prepared rough MnO$_2$ nanomaterial substrates that provide more binding sites for antibody grafting, increase the contact probability between cells and materials and improve the capture efficiency. The highest capture efficiency was 83.2% under the incubation time of 120 min. The MnO$_2$ nanosubstrate was dissolved by employing a $2 \times 10^{-3}$ M concentration of oxalic acid to release the captured cells. The cell release efficiency was up to 91.46% with a reaction time of 60 s. The released cells had a strong ability to proliferate after being collected and re-cultured for 24 h. Identifying and counting CTCs from the peripheral blood of breast cancer patients through the three-color immunocytochemistry method proved the effectiveness of our design platform. Such a simple and economical approach provides a promising platform for the capture and release of cells in clinical research.

**Keywords:** circulating tumor cells; efficient isolation; high activity; convenient platform

## 1. Introduction

Circulating tumor cells (CTCs) are tumor cells that fall off from the surface of the primary tumor. They can invade blood vessels in the human peripheral blood system and participate in human blood circulation [1,2]. CTCs carry a large amount of information, which is considered to be the carrier of cancer metastasis [3,4]. Separating CTCs from peripheral blood is a type of non-invasive tumor "liquid biopsy" [5,6]. This detection method has several advantages, including simple and multiple sampling, high safety and low cost. It is expected to replace the traditional tissue biopsy in monitoring treatment response and determining the prognosis of patients.

In recent years, some sensitive and specific CTC detection techniques have been established, which are broadly divided into physical and chemical methods. The physical method mainly involves filtering technology chip [7,8], lateral displacement technology chip [9,10], spiral structure separating chip [11,12], bulk acoustic wave and surface acoustic wave (SAW) devices [13–15]. They sort tumor cells on the basis of cell size, morphology, density, volume compression ratio and so on. The chemical method achieves tumor cell sorting mostly through antigen–antibody-specific binding. Antibodies [16–18] or Aptamer [19–21] are immobilized on various surfaces, such as microfluidic channels [22], magnetic nanoparticles [23] and microarrays [24], specifically identifying and capturing target cells. However, because of the cellular affinity, the feasible release of captured cancer cells has become a major obstacle to follow-up analysis.

The CTC isolation technique is becoming more advanced with the development of nanomaterials, due to their high specific surface areas, size, shape and unique surface chemical and optical properties. In addition, the structure of nanomaterials is close to the surface structure of cells, which provides more binding sites for antibody grafting and increases the contact probability between cells and materials, thus improving the capture efficiency of cells [25,26]. In the early diagnosis and treatment of cancer, it is very attractive to prepare nanomaterials with different morphologies via controlling the conditions for enhancing the adsorption capacity between materials and cells. Nevertheless, statically captured CTCs bind firmly to the substrate, hindering their release [27,28]. In terms of the molecular analysis of the selected CTCs, it is necessary to develop a better platform that can not only achieve efficient sorting of CTCs, but also keep the activity of the released cells.

In recent years, the research focuses on cancer detection, drug delivery and anti-cancer. Some teams utilized the graphene material system as an ultra-sensitive biosensor for the detection of biological fluids [29], and $Fe_3O_4$ and graphene material system was used for drug delivery [30–34]. In addition, Gholami et al. [35] tried to use 3D nanostructures for cancer treatment, and Ahmadi et al. [36] used the therapy of Ag and magnetic carriers [37] for anticancer research. In addition, some nanomaterials are also widely used in biomedicine [38–42].

In this paper, we designed an ultra-convenient $MnO_2$ nanoparticle substrate for the capture and release of CTCs. The surface of $MnO_2$ is rich in hydroxyl groups, which are easily modified and facilitate the grafting of epithelial cell adhesion factor antibody (anti-EpCAM antibody). In addition, the preparation method of substrates through self-assembling a layer of $MnO_2$ nanoparticles with a hollow structure on transparent glass is simple and cost-effective. The good light transmittance of the substrate is convenient for cell observation at the same time. Furthermore, $MnO_2$ nanoparticles can be dissolved by low-concentration oxalic acid, enabling the high activity of the released cells. This highly transparent and easily dissolved nanoparticles platform provides a new approach for biological research at the cellular and molecular levels.

## 2. Materials and Methods

### 2.1. Fabrication of MnO₂ Nanoparticles

We used a two-step synthesis method to obtain $MnO_2$ nanoparticles. Firstly, 0.641 g $BaMnO_4$ was initially dissolved in 25 mL DI water. Then, 0.67 mL of 4.23 M $H_2SO_4$ was added dropwise to this solution Equation (1) at room temperature. Additionally, the solution was stirred quickly for 10 min. Subsequently, we centrifuged the solution at 1200 $g$ for 3 min. The supernatant was taken out to obtain the desired product-permanganic acid.

$$3BaMnO_4 + 3H_2SO_4 \rightarrow 3BaSO_4 + MnO_2 + 2HMnO_4 + 2H_2O \tag{1}$$

Secondly, the FTO glass was put against the wall of a 100 mL beaker. The resultant permanganic acid was added dropwise to 75 mL of 0.338 g $MnSO_4$ while being rapidly stirred for 15 min at room temperature Equation (2). $MnO_2$ monolayer thin films were obtained, consequently.

$$2HMnO_4 + 2H_2O + 3MnSO_4 \rightarrow 5MnO_2 + 3H_2SO_4 \tag{2}$$

Materials, circulating tumor cell preparation, surface modification, cell capture and release, cell propagation and culture, blood sample treatment and the three-color fluorescence method were described in our previous work [43].

### 2.2. Characterization of MnO₂ Substrate

Figure 1a shows that the surface of the $MnO_2$ nanomaterial contains hydroxyl groups, also facilitating the subsequent modification. The good light transmission was the premise to observe the cell capture effect using a microscope. The UV-Vis absorption spectroscopy (UV-2550, Shimadzu, Kyoto, Japan) test results showed that the self-assembled thin layer of

$MnO_2$ nanoparticles had high light transmittance (Figure 1b). This was also convenient for using an ultra-low concentration of oxalic acid to dissolve the $MnO_2$ nanoparticles, thereby releasing the captured cells.

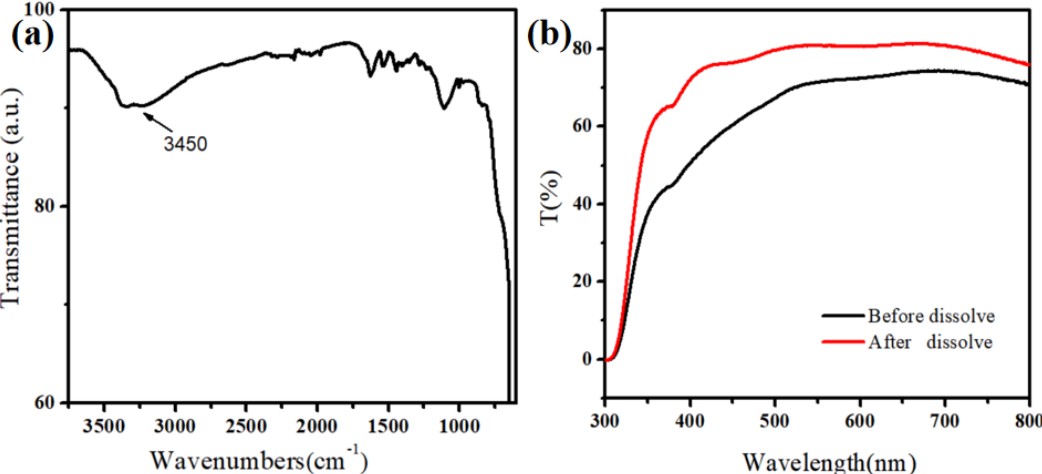

**Figure 1.** (**a**) Infrared spectra of $MnO_2$ nanoparticles. (**b**) Light transmission of $MnO_2$ nanoparticles on the FTO substrate.

The surface roughness and particle sizes of $MnO_2$ were characterized by atomic force microscopy (AFM, Veeco multimode, Amsterdam, The Netherlands). Figure 2 shows the 2D and 3D AFM images of $MnO_2$ substrate (the scan area was $5 \times 5 \ \mu m^2$). The root-mean-square (Sq) roughness of $MnO_2$ is 18.7 nm. The rough interface provides more binding sites for grafted antibodies and increases the contact area between the cells and substrate, which is beneficial to improve the capture efficiency of cells.

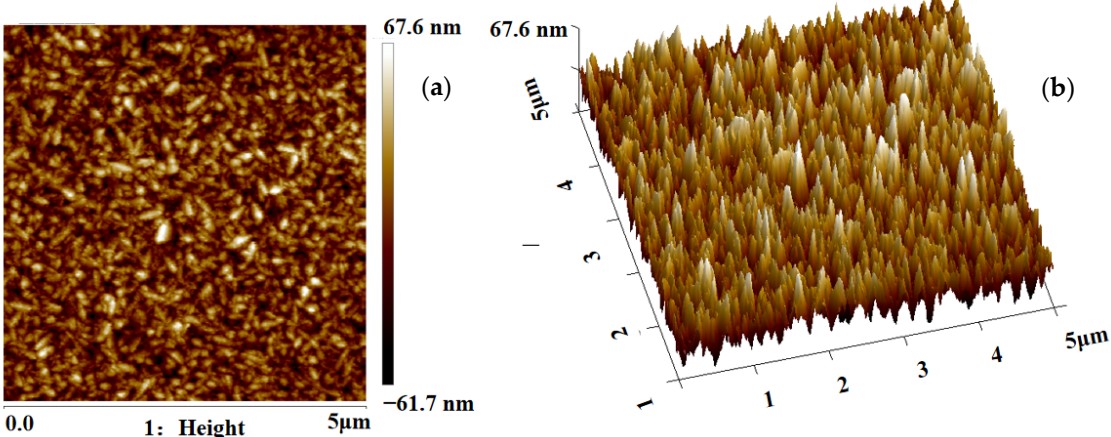

**Figure 2.** (**a**) Two-dimensional AFM images of the $MnO_2$ nanoparticles. (**b**) Three-dimensional AFM images of the $MnO_2$ nanoparticles.

At the same time, we observed the morphology of the substrate with a long self-assemble time. The results are shown in Figure 3. Moreover, the X-ray photoelectron spectroscopy (XPS) characterization of the material (Figure 3) was performed to confirm that Mn existed in the form of $MnO_2$.

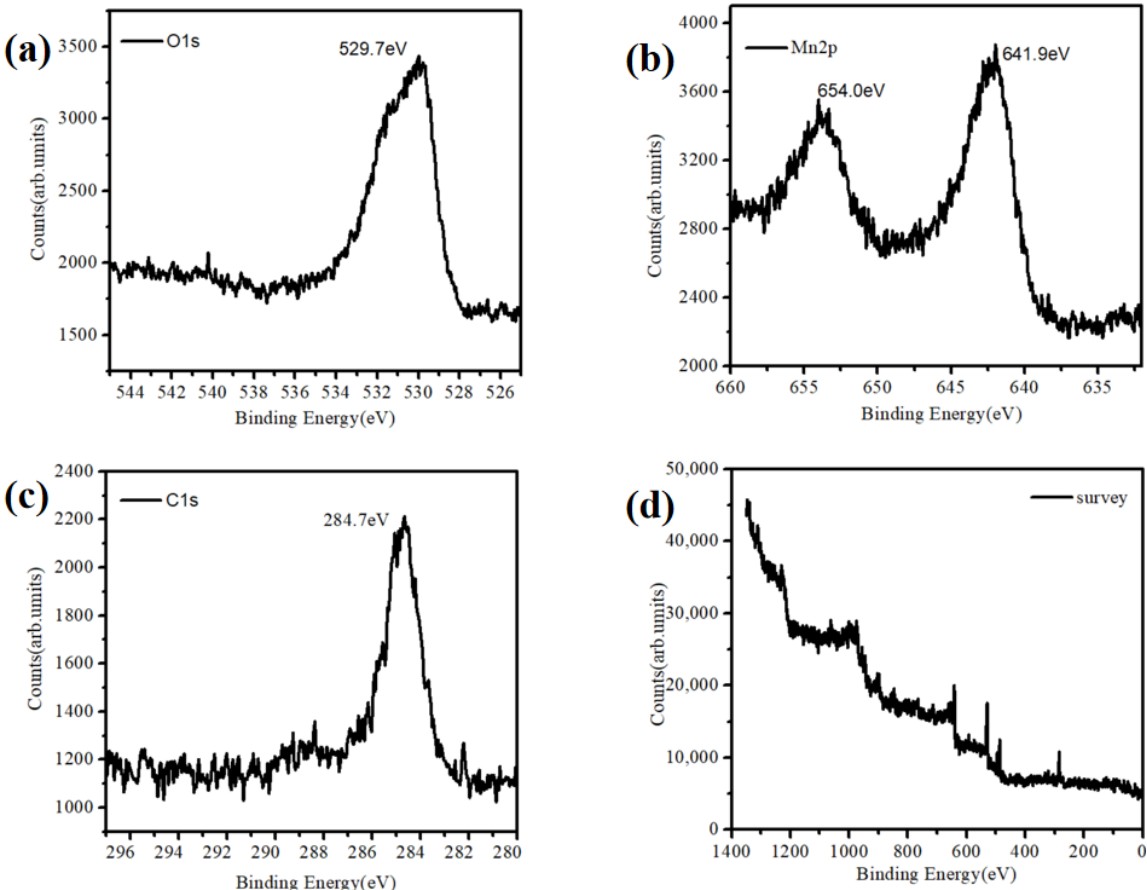

**Figure 3.** XPS characterization of the MnO$_2$ nanoparticles. (**a**) O1s, (**b**) Mn2p, (**c**) C1s and (**d**) survey.

## 3. Results

### 3.1. Surface Modification

Figure 4 is the modification of MnO$_2$ substrate for the capture and release of circulating tumor cells. First, we self-assembled a layer of MnO$_2$ nanoparticles on the FTO glass. After the silane-based reaction and the protein cross-linking reaction, the substrate was attached to biotin-based streptomycin and antibodies (anti-EpCAM) to complete cell capture. Subsequently, the low concentration oxalic acid was used to dissolve the MnO$_2$ nanoparticles and collected the released cells into the centrifuge tube for single-cell analysis.

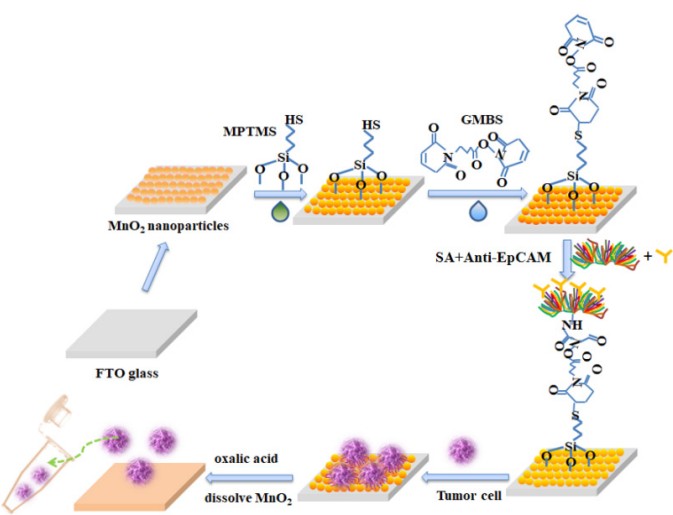

**Figure 4.** The modification of the MnO$_2$ substrate for the capture and release of circulating tumor cells.

### 3.2. Capture and Release of CTCs, Activity and Proliferation

Figure 5a displayed the relationship between the capture efficiency of MCF-7 cell lines and incubation time based on the $MnO_2$ nanoparticle substrate. We cultured the $MnO_2$ substrate under 37 °C and 5% of $CO_2$, and it was taken out at intervals. the suspension cells were washed with PBS. Then, the substrate was placed under a microscope IX71 (Olympus, Tokyo, Japan) to count the captured cells and calculate the capture efficiency. The result presented that the capture efficiency was only 25% when the incubation time was 30 min. Moreover, the capture efficiency gradually increased with the incubation time. When the incubation time reached the optimum of 120 min, the capture efficiency was up to 83.2%.

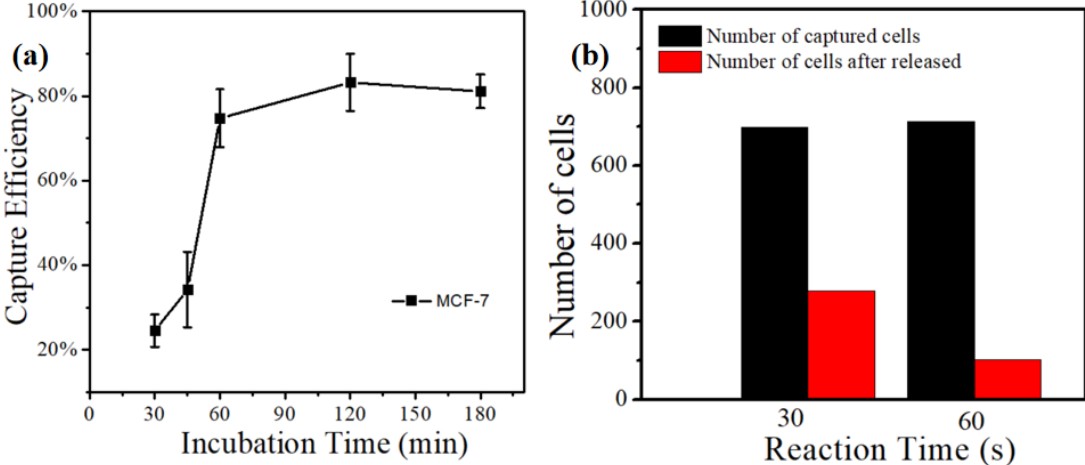

**Figure 5.** (**a**) The relationship between capture efficiency of MCF-7 cell lines and incubation time based on the $MnO_2$ nanoparticle substrate. (**b**) The release effect of MCF-7 cell lines using 0.2 mM concentration of oxalic acid to dissolve $MnO_2$ substrates with different times.

Our previous work [44] indicated that the release efficiency reached up to 89.9% when the concentration of oxalic acid was $2 \times 10^{-3}$ M. In this experiment, we still used the same concentration of oxalic acid to dissolve the $MnO_2$ substrate (Figure 5b). It can be observed that the release efficiency was just 25% with an action time of 30 s. When the action time was extended to 60 s, the release efficiency reached up to 91.46%, Further extending the action time of oxalic acid and substrate did not improve the release efficiency. At the same time, the change in the particles' morphologies on the surface of the substrate can be observed in the FE-SEM images (Figure 6). From this point of view, the release efficiency of the substrate is not only dependent on the concentration of oxalic acid, but also on the action time of oxalic acid and the substrate.

Next, we investigated the effects of nanomaterials on cell capture efficiency. The antibodies on the FTO substrate and $MnO_2$/FTO substrate were modified, respectively. Additionally, the same number of cells (1000 MCF-7 cells) labeled with the FDA were added. The cell capture effect can be viewed directly through a fluorescence microscope. On the $MnO_2$/FTO substrate (Figure 7b), the cell capture efficiency is much higher than that of cells on the FTO substrate (Figure 7a). FE-SEM results showed that the cells on the FTO substrate (Figure 7c) have a spherical shape with a very small amount of outstretched filopodia (Figure 7c), while a large number of outstretched filopodia were extended on the $MnO_2$/FTO substrate (Figure 7d). Combined with the AFM and FE-SEM images of $MnO_2$ nanoparticles, it demonstrated that the nanomaterial increased the roughness of the substrates and the contact areas between the substrates and cells. These results were beneficial to cell creep and migration, thus increasing the capture efficiency. Then, the released cells were collected and cultured for 0 h and 24 h, respectively, in the same environment of 37 °C and 5% $CO_2$. Subsequently, the suspended cells were washed with PBS. As shown in Figure 7e–f, the released cells from the substrate were highly active. After being cultured for 6 h, they grew attached to the plate and their number increased

exponentially. When the incubation time was extended to 24 h, the cell spread more filopodia and covered the entire field of vision. Additionally, it can be observed that the released cells, with a concentration of $2 \times 10^{-3}$ M, have a strong proliferation capacity.

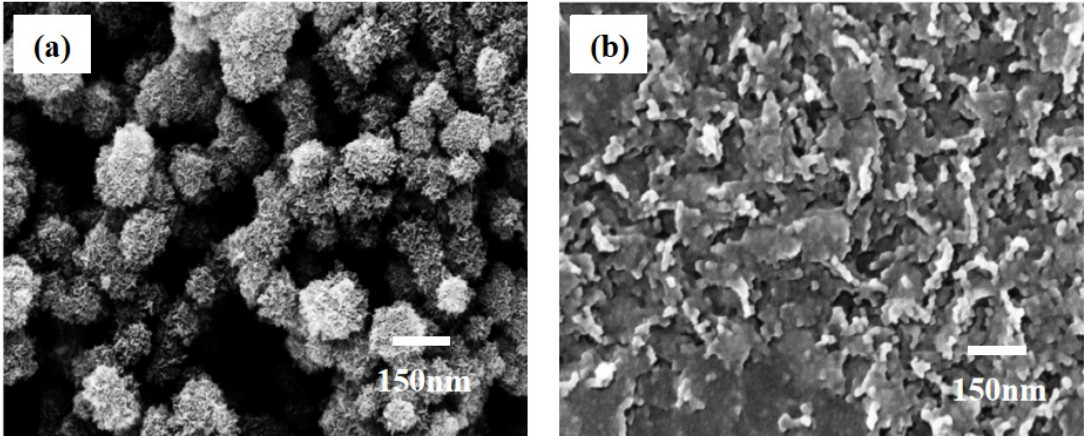

**Figure 6.** Changes in morphology of $MnO_2$ nanoparticles under the $2 \times 10^{-3}$ M concentration of oxalic acid. (**a**) Topography of $MnO_2$ nanoparticles before oxalic acid treatment. (**b**) Topography of $MnO_2$ nanoparticles after oxalic acid etching.

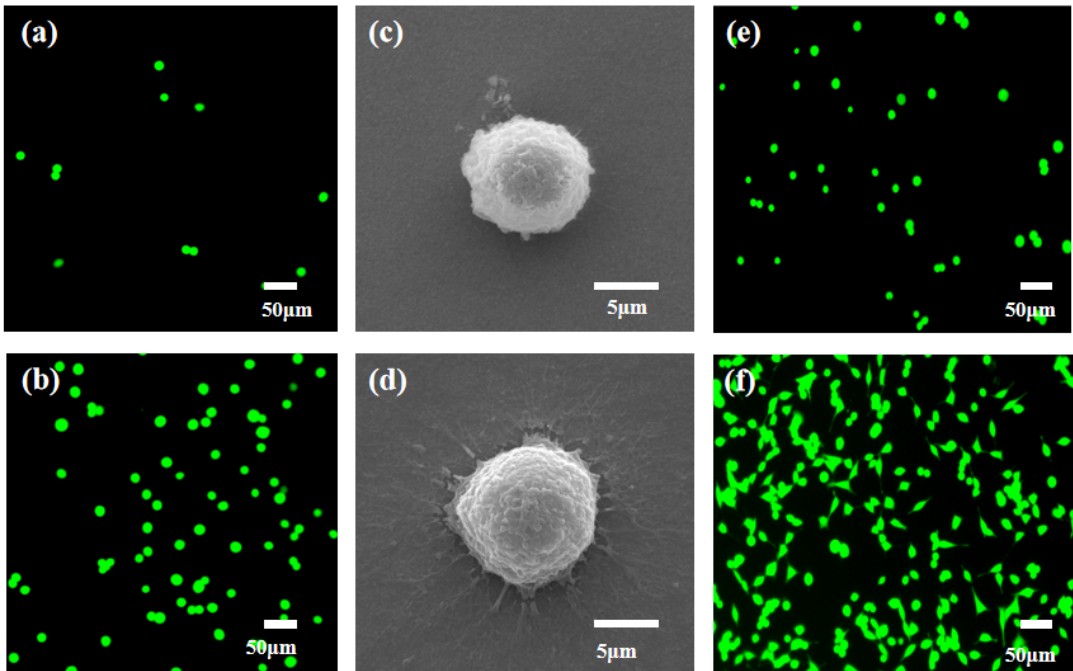

**Figure 7.** Fluorescence images of green-dye-labeling MCF-7 cells on the different substrates. (**a**) FTO substrate and (**b**) $MnO_2$/FTO substrate (Scale bar: 50 μm). (**c,d**) FE-SEM images of captured MCF-7 cells on the different substrates: (**c**) FTO substrate and (**d**) $MnO_2$/FTO substrate (Scale bar: 5 μm). The proliferation ability of released cells from the $MnO_2$ nanosubstrate using 0.2 mM concentration of oxalic acid after being cultured at different times: (**e**) 0 h and (**f**) 24 h (Scale bar: 50 μm).

### 3.3. Identification of Captured Cells from Breast Cancer Patient Blood Samples

Under the optimal experimental conditions, we were able to isolate CTCs from nine cases of breast cancer patient peripheral blood samples. FITC-labeled anti-CD45, PE-labeled anti-cytokeratin and DAPI-labeled cell nuclei were used in our experiments (Figure 8a). The CTCs that had nonspecific adsorption of white blood cells were identified if the cell was presented as CK+/CD45−/DAPI+. On the basis of this principle, we counted

CTCs captured from the peripheral blood of nine cases of breast cancer patients, the sample numbers are ranged from#1 to #9. The number of captured cells ranged from 1 to 12 cells/mL (Figure 8b). This manifested that our $MnO_2$ nanomaterial substrate was an efficient capture and non-destructive release platform for CTC testing in clinical samples with promising applications in the early diagnosis and treatment of cancer patients.

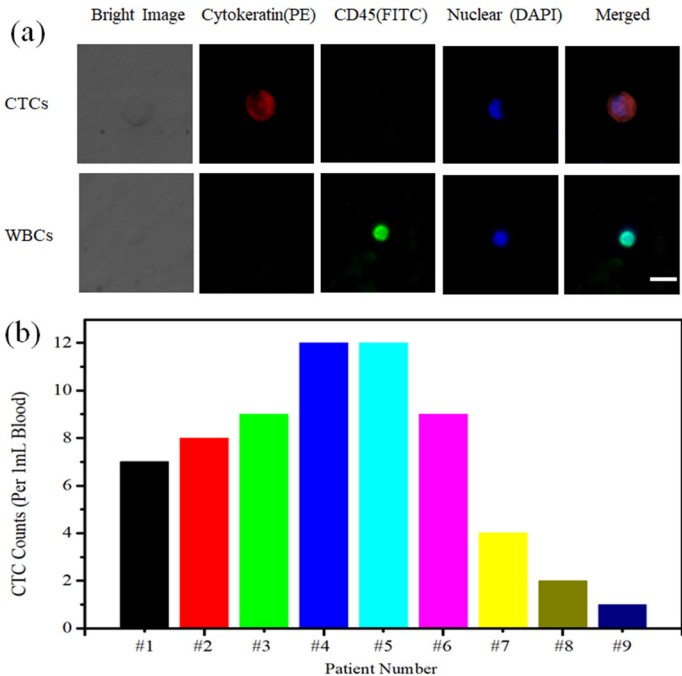

**Figure 8.** (**a**) Fluorescence identification of CTCs captured from breast cancer patient blood samples (Scale bar: 30 μm). (**b**) CTC enumeration captured from 1 mL breast cancer patient blood samples, the sample numbers are ranged from #1 to #9.

## 4. Conclusions

We successfully prepared an ultra-convenient $MnO_2$ nanomaterial substrate and applied it to the capture and release of tumor cells. The rough nanomaterial substrate provided more binding sites for antibody grafting, which was more conducive to cell climbing, thus achieving a higher capture effect. The XRD results showed the very good crystallinity of the nanoparticles and the particles grow in the crystal direction (310), (400), (211), (411), (521) and (002). The FE-SEM and AFM characterization revealed that the nanoparticles were uniform in size and distributed in the range of 150~180 nm. In addition, XPS confirmed the existence of $MnO_2$ nanoparticles on the sample surface. When the incubation time of 120 min, the best capture efficiency is 83.2%. Then, the effects of action time of oxalic acid on the release efficiency and cell activity after release were studied. Finally, the tumor cell sorting in the peripheral blood of patients with breast cancer proved the feasibility of our nanomaterial substrates.

**Author Contributions:** R.L. designed the experiments and carried out cell capture and release experiments. Y.W. and T.L. contributed to experimental data and analysis. The manuscript was written through the contributions of all authors. All authors have read and agreed to the published version of the manuscript.

**Funding:** This work was supported by the Natural Science Foundation of Xinjiang Uygur Autonomous Region of China (Grant No. 2021D01C112), the Doctoral Program of Tian Chi Foundation of Xinjiang Uygur Autonomous Region of China (Grant No. TCBS202042), the Doctoral Program Foundation of Xinjiang University (Grant No. 2020640020) and the Tianshan Innovation Team Program of Xinjiang Uygur Autonomous Region (Grand No. 2020D14038).

**Institutional Review Board Statement:** The study was conducted in accordance with the Declaration of Helsinki, and approved by the Institutional Review Board (or Ethics Committee) of NAME OF INSTITUTE (protocol code 2020173 and 20201018 date of approval).

**Informed Consent Statement:** Informed consent was obtained from all subjects involved in the study.

**Data Availability Statement:** Not applicable.

**Conflicts of Interest:** The authors declare no conflict of interest.

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
