# Peer review of "Efficient Isolation of Circulating Tumor Cells Using Ultra-Convenient Substrates Based on Self-Assembled Hollow MnO2 Nanoparticles"

_coatings, doi:10.3390/coatings12081214_

Round 1
Reviewer 1 Report
The paper reports on using MnO2 nanomaterial substrates for isolation of CTCs for further use. The substrate material is hollow MnO2 nanoparticles, however this was already reported elsewhere for the same purpose: https://pubs.acs.org/doi/abs/10.1021/acsami.8b04683
Therefore, there are novelty issues regarding the planning of this study. Besides, the paper is written in a very amateur way with extremely high number of writing errors and explanation mistakes.
Further, more importantly, I have scientific concern regarding the study, for instance, why did the authors use hollow nanoparticles? There is no reasoning or explanation behind this approach? Why do authors consider the crystallinity of the nanoparticle? How will it help to achieve the end goal in this paper? To me, they seem pointless and gives me the feeling that just to provide more data, such analyses were performed.
The current version of this work is not mature and should not be published.
Author Response
Dear Reviewer:
Thank you for your thoroughly reviewing our manuscript and making many thoughtful comments. I am sorry that my research paper “Design an ultra- convenient substrate based on self-assembled hollow MnO2 nanoparticles for efficient isolating of circulating tumor cells” did not meet your requirements. Here we resubmit a new version of our manuscript, which has been carefully revised according to the reviewers’ comments. The amendments are highlighted in red colour in the revised manuscript, listed one by one below this letter. We hope it will be considered to publish in Coatings. I look forward to learning your response to our submission. Thanks for your time and attention of handling our manuscript.
Comment 1:
The paper reports on using MnO2 nanomaterial substrates for isolation of CTCs for further use. The substrate material is hollow MnO2 nanoparticles, however this was already reported elsewhere for the same purpose: https://pubs.acs.org/doi/abs/10.1021/acsami.8b04683
Therefore, there are novelty issues regarding the planning of this study.
Response 1:
Thank you for your thoroughly reviewing our manuscript. We designed an ultra- convenient substrate based on self-assembled hollow MnO2 nanoparticles for efficient isolating of circulating tumor cells. We have conducted a series of studies on tumor cell isolating,the specific research contents are as follows:
https://pubs.acs.org/doi/abs/10.1021/acsami.8b04683
https://pubs.acs.org/doi/10.1021/acsabm.8b00333
https://www.sciencedirect.com/science/article/abs/pii/S0039914019304850
https://pubs.acs.org/doi/10.1021/acsabm.0c00957
https://iopscience.iop.org/article/10.1088/2399-1984/abbf02/meta
The purpose of this series of work is to explore the most suitable substrate material to obtain the maximum capture efficiency, and expect to have a wide range of applications in the early diagnosis of cancer. The biggest difference between these two papers is that there is no TiO2 nanorod arrays, and only self-assembly method is used to prepare MnO2 nanoparticles, which is simple and time-consuming. In addition, the substrate is more transparent for viewing and is more easily dissolved by low concentrations of oxalic acid, thus releasing captured cells.
Comment 2:
Besides, the paper is written in a very amateur way with extremely high number of writing errors and explanation mistakes.
Response 2:
Thank you for your professional comments. According to your suggestion, I have contacted a native English teacher who is well-versed in English and has experience in writing papers to help me improve the text. The amendments are highlighted in red colour in the revised manuscript.
Comment 3:
Further, more importantly, I have scientific concern regarding the study, for instance, why did the authors use hollow nanoparticles? There is no reasoning or explanation behind this approach? Why do authors consider the crystallinity of the nanoparticle? How will it help to achieve the end goal in this paper? To me, they seem pointless and gives me the feeling that just to provide more data, such analyses were performed.
Response 3:
Thank you for your professional comments. According to your suggestion, the reasons for adopting hollow nanoparticles have been described in detail in the paper.
We used nanoparticles with hollow structures for two main reasons. On the one hand,
hollow MnO2 nanoparticles has great potential for biomedical application by utilizing its properties like facile surface modification, making it feasible to be modified with anti-EpCAM to efficiently capture EpCAM positive cancer cells. On the other hand, the MnO2 thin film was fabricated through in situ self-assembly of MnO2 hollow nanospheres on glass substrate to form monolayer at room temperature. The monolayer exhibits high degree of transparency for cell observation.
Why do authors consider the crystallinity of the nanoparticle?
MnO2 nanomaterials have a complex crystalline structure, the surface morphology, specific surface area and electrical conductivity of nanoparticles with different crystalline structures have obvious differences. XRD is usually used to characterize the crystal structure. In order to verify the hollow structure of MnO2 nanoparticles prepared by self-assembly method, XRD characterization was carried out.
Thanks again for your careful and comprehensive review. Through the supplements and modifications, the quality of the article has been greatly improved. I hope that I can meet the requirements of the journal through this revision. Thank you very much!

Reviewer 2 Report
About this Design an ultra-convenient substrate based on self-assembled hollow MnO2 nanoparticles for efficient isolating of circulating tumor cells, First of all, the review is specific enough. Authors provide a large amount of general information which the readers can easily obtain from textbooks or other references. The authors, however, did even provide an insight into why there is a great challenge towards the commercialization of this kind of Important role of advanced MnO2 nanoparticles materials. But the key questions are which type(s) of this material are the closest to the commercialization level and why? What are the industry's expectations for this kind of isolating of circulating tumor cells, to be commercialized in terms of performance? Besides, the title of the paper is confusing. In fact, biocompatibility materials are one of the main issues for biomedical activity as it needs to be subject to frequent biocompatibility to recover their materials.
1) Authors should carefully revise and corrected all the grammatical issues and follow the scientific norms in the whole manuscript.
2) Please use updated and recent papers in the literature review to give more sense to the reader.
so, below papers are add to manuscript Recent publications
1: Bio-enhanced polyrhodanine/graphene Oxide/Fe3O4 nanocomposite with kombucha solvent supernatant as ultra-sensitive biosensor for detection of doxorubicin hydrochloride in biological fluids
2: Introduction of magnetic and supermagnetic nanoparticles in new approach of targeting drug delivery and cancer therapy application
3: A conceptual review of rhodanine: current applications of antiviral drugs, anticancer and antimicrobial activities
4: Graphene nano-ribbon based high potential and efficiency for DNA, cancer therapy and drug delivery applications
5: Nanosensors for chemical and biological and medical applications
6: Development of Graphene based Nanocomposites Towards Medical and Biological Applications
7: Recent progress in electrochemical detection of human papillomavirus (HPV) via graphene-based nanosensors
8: Bioactive Graphene Quantum Dots Based Polymer Composite for Biomedical Applications
9: Renewable Carbon Nano-materials: Novel Resources for Dental Tissue Engineering
10: Gold nanostars-diagnosis, bioimaging and biomedical applications
11: Data on cytotoxic and antibacterial activity of synthesized Fe3O4 nanoparticles using Malva sylvestris
12: 3D Nanostructures for Tissue Engineering, Cancer Therapy, and Gene Delivery
13: Green Synthesis of Supermagnetic Fe3O4-MgO Nanoparticles via Nutmeg Essential Oil Toward Superior Anti-Bacterial and Anti-Fungal Performance
14: Asymmetric membranes: A potential scaffold for wound healing applications
15: Anti-bacterial/fungal and anti-cancer performance of green synthesized Ag nanoparticles using summer savory extract
16: Green Synthesis of Magnetic Nanoparticles Using Satureja hortensis Essential Oil toward Superior Antibacterial/Fungal and Anticancer Performance
Author Response
Dear Reviewer:
Thank you for your thoroughly reviewing our manuscript and making many thoughtful comments. The comments are very important for improving our paper, according to your comments, I revised our manuscript carefully. The amendments are highlighted in red colour in the revised manuscript, and listed one by one below this letter. We hope it will be considered to publish in Coatings without any special request. Thank you very much!
Comment 1:
The authors, however, did even provide an insight into why there is a great challenge towards the commercialization of this kind of Important role of advanced MnO2 nanoparticles materials. But the key questions are which type(s) of this material are the closest to the commercialization level and why?
Response 1:
Thank you for your professional comments. Due to some characteristics of CTCs, such as rarity, heterogeneity, cell morphology atypicality, morphology and category uncertainty, which making clinical testing face great challenges. At present, the detection of circulating tumor cells in peripheral blood of patients based on nanomaterials is still in the experimental stage, the detail reasons as follows: Firstly, compared with blood cells, the number of circulating tumor cells in peripheral blood is less and non-specific adsorption is serious. Secondly, the detection sensitivity of nanomaterial based chip is not enough. Thirdly, it is difficult to release captured cells from nanomaterials.
Comment 2:
What are the industry's expectations for this kind of isolating of circulating tumor cells, to be commercialized in terms of performance?
Response 2:
In terms of industrial production, many researchers have been committed to the development of detection chip, and strive to achieve low detection cost, high sensitivity, fast detection speed, suitable for mass production, products are easy to be preserved.
Comment 3:
Authors should carefully revise and corrected all the grammatical issues and follow the scientific norms in the whole manuscript.
Response 3:
Thank you for your professional comments. According to your suggestion, I have contacted a native English teacher who is well-versed in English and has experience in writing papers to help me improve the text. The amendments are highlighted in red colour in the revised manuscript.
Comment 4:
Please use updated and recent papers in the literature review to give more sense to the reader. so, below papers are add to manuscript Recent publications.
Response 4:
Thank you for your recommendation. I have studied the relevant content carefully and have cited the work in the latest manuscript. These works are very excellent and make my articles more comprehensive and more convenient for readers to understand. Thanks again for your recommendation.
Thanks again for your careful and comprehensive review. Through the supplements and modifications, the quality of the article has been greatly improved. We hope it will be considered to publish in Coatings without any special request. Thank you very much! Please see the attachment.

Reviewer 3 Report
This manuscript presents an elegant approach towards circulating cancer cells capture and release for analysis, based on a sacrificial, specifically bio-functionalized MnO2 NP surface coating. The work is timely since circulating cancer cell capture and detection are essential for the improvement of cancer diagnosis approaches. The manuscript should be carefully revised by a native or fluent English speaker.
Further comments and questions:
Please clearly cite, where mentioned, the previous work in which many of the protocols were described. Key parameters should still be given in the current manuscript.
With respect to this previous work, the novelty of the current work should be stressed and demonstrated.
How do oxalic acid concentrations used in this work compare with potentially cytotoxic doses?
Font sizes should be increased, and/or a scale bar added in figure 2 to make it more readable.
Please double-check the scale bars in figure 6, the resolution seems surprisingly high for SEM, in particular with respect to the level of detail in panel (a). Are the particle sizes and roughness matching between SEM and AFM?
The proportion of specific CTC capture with respect to non-specific capture of other cells should be indicated and discussed. How does this proportion compare between standard glass substrates and the new coating?
“Pseudo-foot” should be replaced in the manuscript by more specific cell biology terms, i.e. filopodia, lamellipodia, etc. depending on the pseudopodia described.
Author Response
Dear Reviewer:
Thank you for your thoroughly reviewing our manuscript and making many thoughtful comments. The comments are very important for improving our paper, according to your comments, I revised our manuscript carefully. The amendments are highlighted in red colour in the revised manuscript, and listed one by one below this letter. We hope it will be considered to publish in Coatings without any special request. Thank you very much!
Comment 1:
1) Please clearly cite, where mentioned, the previous work in which many of the protocols were described. Key parameters should still be given in the current manuscript.
Response 1:
Thank you. According to your suggestion. I added the corresponding key parameters to the current manuscript and refined the references. Thank you again for your professional review.
Comment 2:
With respect to this previous work, the novelty of the current work should be stressed and demonstrated.
Response 2:
Thank you for your professional comments. With respect to this previous work, the novelty of the current work is reflected in three aspects: Firstly, hollow MnO2 nanoparticles has great potential for biomedical application by utilizing its properties like facile surface modification, making it feasible to be modified with anti-EpCAM to efficiently capture EpCAM positive cancer cells. Secondly, the MnO2 thin film was fabricated through in situ self-assembly of MnO2 hollow nanospheres on glass substrate to form monolayer at room temperature. The monolayer exhibits high degree of transparency for cell observation. Thirdly, the MnO2 film can be easily dissolved by oxalic acid with extreme low concentration. This facile room temperature reaction offers the feasibility that could release the captured cells with high viability.
Comment 3:
How do oxalic acid concentrations used in this work compare with potentially cytotoxic doses?
Response 3:
Thank you for your professional comments. I'm sorry that the potential cytotoxicity measurement was not studied in this work. The teacher's question also provided us with a new scientific research idea for the following research. In the following work, we must take into account the relationship between oxalate concentration and potential cytotoxic dose in accordance with the teacher's suggestion.
Comment 4:
Font sizes should be increased, and/or a scale bar added in figure 2 to make it more readable.
Response 4:
Thank you for your professional comments. I have increased the font sizes and added a scale bar in Figure 2.
Comment 5:
Please double-check the scale bars in figure 6, the resolution seems surprisingly high for SEM, in particular with respect to the level of detail in panel (a). Are the particle sizes and roughness matching between SEM and AFM?
Response 5:
Thank you for your professional comments. After several confirmations, the scale bars with no problem. The main reasons for the large difference in size and roughness between SEM and AFM are as follows. AFM is the characterization of MnO2 substrate material that captures the substrate, so as to observe the morphology and roughness of the substrate. In order to describe the morphology of the substrate more clearly, SEM deliberately uses the substrate materials with longer growth time to characterize the differences in morphology before and after oxalic acid dissolution. I'm sorry for causing trouble to the teacher's understanding.
Comment 6:
The proportion of specific CTC capture with respect to non-specific capture of other cells should be indicated and discussed. How does this proportion compare between standard glass substrates and the new coating?
Response 6:
Thank you for your professional comments. According to your suggestion, we combed and calculated the original data again, and the results shows that the proportion of specific CTC capture with respect to non-specific capture of other cells is 8.94:1. we need to continuously optimize experimental parameters to effectively reduce the non-specific adsorption of blood cells.
Comment 7:
“Pseudo-foot” should be replaced in the manuscript by more specific cell biology terms, i.e. filopodia, lamellipodia, etc. depending on the pseudopodia described.
Response 7:
Thank you for your professional comments.I am very sorry for my lack of professionalism and made a mistake. According to your suggestion, I have changed the pseudo-foot in the manuscript to filopodia. Next, I will study hard to improve my professional level.
Thanks again for your careful and comprehensive review. Through the supplements and modifications, the quality of the article has been greatly improved. We hope it will be considered to publish in Coatings without any special request. Thank you very much! Please see the attachment.

Round 2
Reviewer 1 Report
The manuscript has improved to some extent, however there are still unresolved issues. For instance, there are still substantial grammatical and writing errors. I am not sure if this draft went through a native speaker edit. I am providing a few examples below where there are issues:
Line 14: Using self-assembly method, we prepare rough MnO2 ...
Lines 156-157: At the same time, the change of particles’ morphology on the surface of the substrate can be observed in the FE-FE-SEM images (Figure 6).
Lines 185-186: When the incubation time was further extend to 24 h, the cell spread morefilopodia185 and covered the entire field of vision.
Figure 4 has text on it with underlines. This must be removed and corrected.
Regarding the crystallinity of the hollow MnO2 nanoparticles, the authors replied as: "MnO2 nanomaterials have a complex crystalline structure, the surface morphology, specific surface area and electrical conductivity of nanoparticles with different crystalline structures have obvious differences. XRD is usually used to characterize the crystal structure. In order to verify the hollow structure of MnO2 nanoparticles prepared by self-assembly method, XRD characterization was carried out."
However, the crystal structure has nothing to do with the CTC capture. The CTC capture is a surface phenomenon. How does crystallinity affect the capture process? I cannot see a relation. This must be clarified. Besides, NPs are around 100-150 nm where CTCs are more than 5 um. The hollow structure does not seem to have an effect in the CTC capture process. If it has, it must be clearly explained.
Author Response
Responds to the reviewers’ comments:
Dear Reviewer:
Thank you again for your thoroughly reviewing our manuscript and making many thoughtful comments. The comments are very important for improving our paper, according to your comments, I revised our manuscript carefully. Here we resubmit a new version of our manuscript, which has been carefully revised according to your comments. The amendments are highlighted in red colour in the revised manuscript, listed one by one below this letter. We hope it will be considered to publish in Coatings. I look forward to learning your response to our submission. Thanks for your time and attention of handling our manuscript.
Comment 1:
The manuscript has improved to some extent, however there are still unresolved issues. For instance, there are still substantial grammatical and writing errors. I am not sure if this draft went through a native speaker edit. I am providing a few examples below where there are issues:
Line 14: Using self-assembly method, we prepare rough MnO2 ...
Lines 156-157: At the same time, the change of particles’ morphology on the surface of the substrate can be observed in the FE-SEM images (Figure 6).
Lines 185-186: When the incubation time was further extend to 24 h, the cell spread morefilopodia and covered the entire field of vision.
Response 1:
Thank you for your professional comments. I am sorry that there are still some grammatical and writing errors after a revision. According to your suggestion, once again invite a professional English teacher who is well-versed in English and has experience in writing papers to help me improve the text. I hoped that after this revision, these errors can be eliminated to meet the requirements of the journal. The amendments are highlighted in red colour in the revised manuscript. Thanks again for your review!
Comment 2:
Figure 4 has text on it with underlines. This must be removed and corrected.
Response 2:
Thank you for your professional comments. I'm sorry for not checking it clearly. According to your requirements, the underline in Figure 4 has been removed. Thanks again for your review!
Comment 3:
Regarding the crystallinity of the hollow MnO2 nanoparticles, the authors replied as: "MnO2 nanomaterials have a complex crystalline structure, the surface morphology, specific surface area and electrical conductivity of nanoparticles with different crystalline structures have obvious differences. XRD is usually used to characterize the crystal structure. In order to verify the hollow structure of MnO2 nanoparticles prepared by self-assembly method, XRD characterization was carried out."
However, the crystal structure has nothing to do with the CTC capture. The CTC capture is a surface phenomenon. How does crystallinity affect the capture process? I cannot see a relation. This must be clarified. Besides, NPs are around 100-150 nm where CTCs are more than 5 um. The hollow structure does not seem to have an effect in the CTC capture process. If it has, it must be clearly explained.
Response 3:
Thank you for your professional comments. When you first mentioned that the crystal structure had nothing to do with the CTC capture,I always felt that the morphology of the material was closely related to its crystal structure, so I tried to explain the necessity of XRD characterization with the corresponding relationship between the structure and morphology. When you mentioned this issue again,I thought it over and over and confirmed that you were right. Besides, you mentioned that NPs are around 100-150 nm where CTCs are more than 5 um. The hollow structure does not seem to have an effect in the CTC capture process. In my opinion, the hollow structure of the MnO2 nanoparticles has a rough surface morphology which can provide more binding sites for antibody, increase the contact area, thus improving the capture efficiency of cells. Based on your question, I agree with your suggestion that the CTC capture is a surface phenomenon. We only need to use SEM to observe the surface morphology of the material, and do not need to consider the crystallinity of the nanoparticle. Therefore, according to your suggestion, I modified Figure 1 and deleted XRD and SEM in Figure 1.
Thanks again for your careful and comprehensive review. Through the supplements and modifications, the quality of the article has been greatly improved. I hope that I can meet the requirements of the journal through this revision. Thank you very much!

This manuscript is a resubmission of an earlier submission. The following is a list of the peer review reports and author responses from that submission.